# Interpretable Transformers by Condition Guided Self-Attention

## Abstract

Transformer models have achieved remarkable success across various domains, sparking breakthroughs in fields beyond their original applications. As a use case from neuroscience, distinguishing Alzheimer's disease (AD) from healthy brain cells using high-dimensional single-cell transcriptomic data is a challenging classification task. While attention-based models offer strong discriminative performance, accuracy alone is not sufficient. It is essential to balance predictive accuracy with biologically meaningful interpretations aligned with domain insights. In this work, we propose an interpretable Transformer architecture based on a Disease-Specific Conditional Guided Self-Attention (DSCGA) mechanism for Alzheimer's disease cell classification. The proposed approach maps each cell gene expression vector into a set of tokens corresponding to a catalogue of known biological pathways. It first learns pathway-based representations to classify cells, followed by biologically guided refinement using our proposed DSCGA mechanism, which amplifies attention to pathways relevant to a specific condition while down-weighting irrelevant ones, adapting to the context of each cell observation Specifically, it extends the standard self-attention mechanism by progressively incorporating a second term that is activated dynamically to help the model focus on condition-related biological pathways. The final attention scores are computed by adding the original self-attention scores and the condition-specific scores. The model requires post-training with DSCGA to condition attention on disease-specific signals. The ultimate goal is to distinguish between AD and healthy cells while generating interpretation-driven predictions aligned with prior biological knowledge. Extensive experiments were carried out using two different real-world datasets, namely Seattle and ROSMAP. Experimental results demonstrate the effectiveness of our proposal and prove its ability to outperform baselines in terms of biological interpretation quality while maintaining a controlled accuracy drop. Precisely, for AD-predicted cells, our method increases the number of correctly identified AD-related pathways using attention scores, from 3.96 to 18.98 (KEGG) and 8.29 to 30.53 (WikiPathways) on Seattle, and from 3.88 to 18.89 (KEGG) and 7.65 to 30.90 (WikiPathways) on ROSMAP. Furthermore, our proposal can be adapted to improve the domain specific interpretability of several existing attention-based architectures if external established knowledge is available.

## 1 Introduction

While attention-based architectures have achieved success in various domains, including neuroscience (Hao et al., 2024; Chen et al., 2023; Cui et al., 2024; Theodoris et al., 2023; Yang et al., 2022), their interpretability remains a significant challenge. A typical transformer model comprises multiple blocks, which include self-attention, normalization, and feed-forward network. These blocks capture complex relationships within biological sequential data but do so in ways that are often difficult to interpret (Vaswani, 2017; Touvron et al., 2023a;b; Chen et al., 2021; Devlin et al., 2018).

Alzheimer's disease (AD) is a prevalent neurodegenerative pathology, which is deemed a form of dementia and a major contributor to disability among elderly individuals (Hodgson et al., 2024; Botto et al., 2022; Winblad et al., 2016; Singh et al., 2024). Recent advancements in the neuroscience field have shed light on the molecular and cellular pathways involved, opening the door to the development of powerful tools and therapeutic interventions aimed at slowing the disease. In particular, research on AD seeks to uncover the underlying factors and genetic influences contributing to multiple risks, develop effective treatments to combat this challenging disease and provide high-quality patient care (Zhang et al., 2024a; Lambert et al., 2023; Hebert et al., 2010; Dubois et al., 2010; James et al., 2014; Ferreira et al., 2020).

In particular, the nature of attention mechanisms in transformer models, coupled with the non-linearity, makes it challenging to understand why this model makes a specific prediction, especially in applications where explainability is important such as detecting Alzheimer's disease or healthy cells, as accurate predictions alone are often insufficient for the goal of studying biology. Understanding the biological reasoning behind these predictions is crucial. However, the standard attention mechanisms often fail to highlight biologically meaningful relationships between relevant elements.

In previous research, several works (Theodoris et al., 2023; Cui et al., 2024; Yang et al., 2022; Chen et al., 2023) present models with high discriminative capacity, but their lack of biological interpretability hinders trust in their predictions. This study focuses on improving the biological interpretability of a Transformer model trained to classify Alzheimer's disease (AD) and healthy (control) from single cell transcriptomic datasets. By explicitly integrating external biological knowledge into the attention mechanism, we demonstrate a proof of principle using single-cell transcriptomics data, which represents high-dimensional, challenging to model biological measurements. Our approach aims to enhance the model's interpretability, enabling more meaningful results and deeper insights into disease mechanisms.

In this paper, we adapted the LLaMA model's transformer block (Touvron et al., 2023a;b) due to several factors, including the nature of the problem, the data used, quality of predictions, and our need to train from scratch. We designed an alternative to the standard self-attention mechanism, termed disease-specific condition guided self-attention (DSCGA), tailored to replace the one used in many existing works (Theodoris et al., 2023; Yang et al., 2022; Chen et al., 2023), and enhance the biological interpretability of Attention-based architectures for cell disease classification, and other application areas with external reference knowledge. Increasing accuracy is beyond the scope of this work, as it has already been addressed in numerous existing studies. The contributions of our present work are summarized as follows:

- We proposed an effective technique called disease-specific conditional guided self-attention to improve the overall biological interpretability of transformers for single cell disease classification.
- We devised a conditional layer that takes advantage of dynamic selection strategy according to the provided model's outcome for improving the biological interpretability.
- We developed a strategy that adopts cell sampling to fine-tune only the important network layers relevant to each predicted condition and yields high-quality biological interpretations of the conditions of interest.

## 2 RELATED WORK

This section covers a set of state-of-the-art Transformer architectures based on attention mechanism, developed for single-cell data analysis. For instance, Geneformer (Theodoris et al., 2023) is a transformer architecture designed to handle transcriptomic data, trained on a large dataset of 30 million cells. As a context-aware model, it adjusts its understanding of each gene based on the specific characteristics of the cell, such as cell type, disease state, or developmental stage. The model employs self-attention as a key component in each block, enabling it to enhance prediction quality in settings with limited biological data. Similarly, ScGPT (Cui et al., 2024) is a pre-trained generative model tailored to process single-cell RNA sequencing (scRNA-seq) data. It can be applied to various downstream applications,

such as cell type annotation, gene network inference, and perturbation response prediction. This model adopts attention mechanisms to generate contextualized vector embeddings. In addition, scBERT (Yang et al., 2022) is an adapted version of the BERT approach (Devlin et al., 2018), designed for predicting cell types and annotating single-cell RNA-seq data. It can be used for understanding disease progression. Moreover, CellPLM (Wen et al., 2024) is a pre-trained model that handles cells and tissues as a set of tokens and sentences respectively. It encodes cell-cell communication, enhancing its capability for various downstream biological applications. In parallel, TOSICA (Chen et al., 2023) is a transformer-based approach that relies on multi-head self-attention to provide biologically interpretable insights, helping to comprehend cellular behavior during development and disease progression.

Another notable model is xTrimoscFoundation (Hao et al., 2024), a pre-trained foundation model designed to unravel the language of cells. It can capture complex contextual relationships among genes across diverse cell types. Meanwhile, ACTINN (Ma & Pellegrini, 2020) is a three-layer neural network architecture developed for automated cell type classification. It is trained on datasets with annotated cell types and is then applied to predict cell types in unseen datasets. Lastly, Cell2Sentence (C2S) (Levine et al., 2023) aims to adapt large language models (LLMs) to transcriptomic data. It converts single-cell gene expression data into textual sequences and ranks genes based on their expression levels in descending order. This approach enables LLMs to handle biological information while preserving the complexity and richness of single-cell data.

Transformer-based architectures, such as Geneformer, scBERT, TOSICA, and scGPT, rely on attention mechanisms, with scGPT adopting an adapted form. These models are designed to improve the discriminative capacity for various tasks. However, they fall short when it comes to biological interpretability. Specifically, the interpretation of attention score matrices in these models is often inconsistent with biological reasoning. This inconsistency presents a significant challenge. Relying solely on post-hoc 'post mortem' explanations is inadequate for ensuring biological interpretability. In biological contexts, such as transcriptomics, model explanations benefit from direct contextualization with established biological processes. While these models excel in prediction, they lack the transparency and grounding in biology necessary for revealing biologically interpretable insights.

TOSICA, for example, incorporates biological pathways as input and uses a standard self-attention mechanism. Yet, our study demonstrates that conventional self-attention alone is insufficient to capture all the biological nuances associated with different predicted conditions. Moreover, none of the existing models, including scGPT, scBERT, and Geneformer, were designed to directly integrate external biological knowledge or enhance biological interpretability through the attention layer. This represents a critical gap, as incorporating knowledge such as known pathways, gene-disease associations, or other biological frameworks could substantially improve the biological relevance of these models. By integrating this external biological knowledge, we aim to enhance the model's biological interpretability, yielding more grounded predictions anchored in established biological mechanisms.

In this work, we address the challenge of enhancing biological interpretability in transformer models. Our approach focuses on improving the model's understanding of why specific predictions are made and ensuring consistency with known biological knowledge. By incorporating external biological knowledge into the model's architecture, we improve both the accuracy and biological interpretability of the predictions. Specifically, we propose a comprehensive guide for extending a transformer model to enhance its interpretability using our alternative to the standard self-attention mechanism. Analyzing the attention score matrices allows us to gain valuable insights into the model's decision-making process, revealing the biological pathways associated with each predicted condition and improving the biological understanding of the model's predictions.

## 3 Transformer-based Disease-Specific Condition Guided Self-Attention (DSCGA)

We propose Disease-Specific Condition Guided Self-Attention (DSCGA) to address the limitations of standard attention in capturing biologically relevant interactions. Unlike

conventional self-attention, DSCGA incorporates external biological knowledge to improve interpretability and align model predictions with known disease-specific pathways. By modifying the attention mechanism, DSCGA allows the Transformer to highlight biologically meaningful interactions for each predicted condition.

### 3.1 Architectural Modification: Integrating Submodules

Given the pre-trained model denoted $Model_{SA}$, this step aims to modify its architecture by integrating two dedicated submodules. The first replaces the conventional self-attention layer with our proposed DSCGA attention. In particular, it reuses the shared weights extracted from the attention mechanism of the trained model $Model_{SA}$ and introduces new layers. The second module is appended as a secondary output to produce auxiliary information that guides condition-awareness. Figure 1 illustrates the exact architectural modification required for $Model_{SA}$. The resulting architecture, denoted as $Model_{DSCGA}$, integrates the DSCGA self-attention layer.

### 3.2 Disease-Specific Condition Guided Self-Attention (DSCGA)

The DSCGA attention mechanism takes as input $(N+1)$ token vectors $E_i = (E_i^1, \ldots, E_i^{(N+1)})$ and a condition scalar $C_i$. Each token corresponds to a biological pathway, obtained by representing the expression of its member genes from scRNA-seq data (raw count matrix) using external knowledge databases, namely KEGG and WikiPathways. Additional details on the definition of these input vectors are provided in Appendix A.

Let $D_k$ be the query/key embedding dimension, DSCGA maps $E_i$ into two queries, $Q_1 \in \mathbb{R}^{(N+1) \times (D_k)}$ and $Q_2 \in \mathbb{R}^{(N+1) \times D_k}$, two keys, $K_1 \in \mathbb{R}^{(N+1) \times D_k}$ and $K_2 \in \mathbb{R}^{(N+1) \times D_k}$, and a value matrix $V$. A condition-aware attention score matrix $Att_{matrix}$ is computed using two attention matrices influenced by a condition. The first matrix based on $Q_1 \in \mathbb{R}^{(N+1) \times D_k}$ and $K_1 \in \mathbb{R}^{(N+1) \times D_k}$, is calculated using the shared weights from standard self-attention. The second matrix, derived from $Q_2 \in \mathbb{R}^{(N+1) \times D_k}$ and $K_2 \in \mathbb{R}^{(N+1) \times D_k}$, is computed using newly introduced layers. A key design feature is the use of two separate attention matrices, where the second is a condition-dependent matrix that is dynamically activated or deactivated based on the input context. This is followed by a weighted summation of the value vectors. Figure 2 illustrates the proposed DSCGA attention mechanism formulated as follows:

$$DSCGA(E_i, C_i) = (Att_{matrix} \circ Repeat(Agg(Att_{matrix})W_{agg}))V$$
$$Att_{matrix} = Softmax(F_{base}(Q_1, K_1) + F_{condition}(C_i, Q_2, K_2)) \tag{1}$$

Here, $Softmax(.)$ represents the softmax function. $DSCGA(\cdot)$ refers to our proposed condition-guided attention mechanism. The symbol $\circ$ denotes the Hadamard product (element-wise multiplication). $Agg(\cdot)$ is a function that aggregates the attention matrix across tokens, yielding a summary of token importance for the full input under the current condition. $W_{agg} \in \mathbb{R}^{(N+1) \times (N+1)}$ denotes a learnable weight matrix applied to the aggregated vector, allowing the model to reweight the global importance signal. $Repeat(\cdot)$ is a function that takes the aggregated importance vector and repeats it along the token dimension to match the shape of the attention matrix denoted $Att_{matrix}$. This latter is computed using two functions, $F_{base}(.)$ for the global attention scores and $F_{condition}(.)$ producing conditioned attention scores dependent on the input context. Note that $Att_{matrix}$ is estimated progressively, following the progressive training step described in the next section.

### 3.3 Progressive Post-training for Biological Interpretability

This section covers the progressive post-training phase of $Model_{DSCGA}$ based on our Disease-Specific Condition Guided Self-Attention (DSCGA). Note that the components of the architecture are added progressively as indicated throughout the post-training stages.

#### 3.3.1 Data sampling and conditional guidance of the attention layer

Given the trained architecture $Model_{SA}$, this step involves incorporating a second term into the standard self-attention mechanism using additional layers. This process is defined as

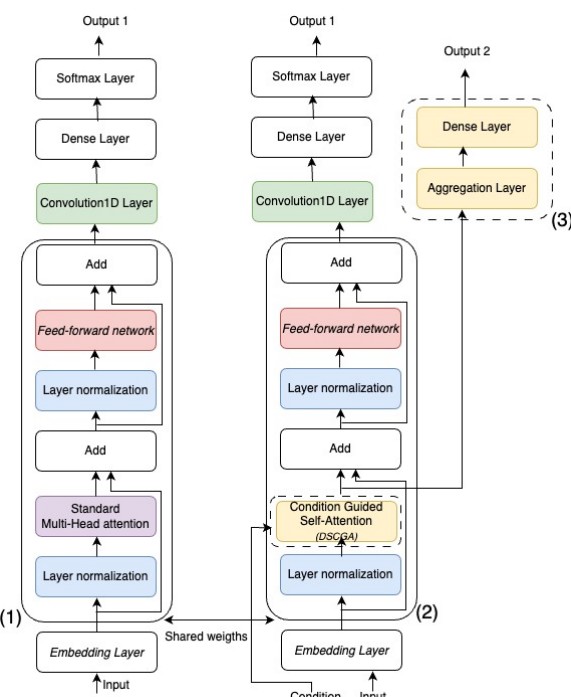

Figure 1: (1) The core transformer architecture based on standard self-attention ($Model_{SA}$). (2) The interpretable architecture obtained by replacing self-attention with DSGCA ($Model_{DSCGA}$). (3) The submodule added as a second output to guide biological learning.

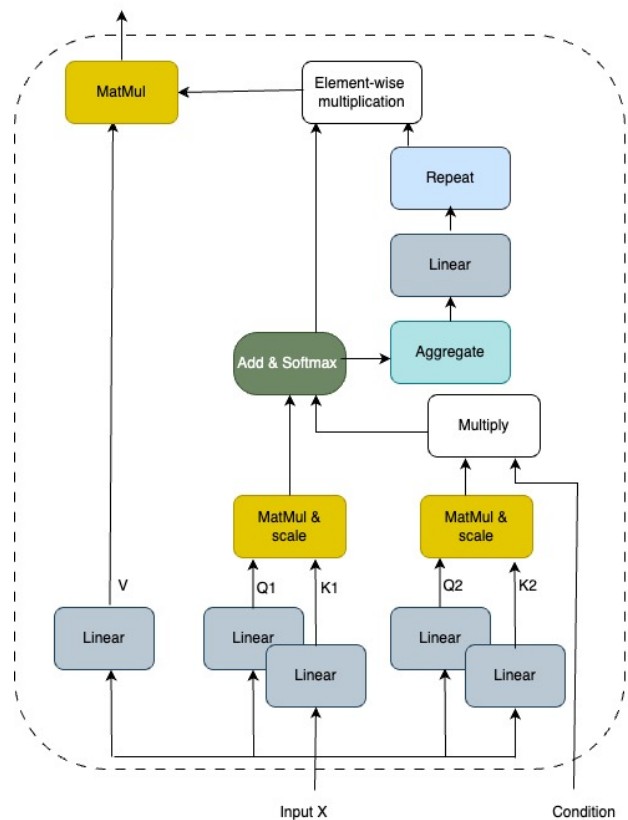

Figure 2: Disease-Specific Condition Guided Self-Attention (DSCGA)

follows:

$$\text{Softmax}\left(\frac{M_A + M_{A2}}{\sqrt{d_k}}\right)V \tag{2}$$

$$M_{A2} = Q_2 K_2^T ; Q_2 = X W_2^q ; K_2 = X W_2^k ;$$

Where $M_{A2}$ is a second attention matrix designed to capture relevant biological information. $W_2^q$ and $W_2^k$ are newly introduced weight matrices added to the trained model $Model_{SA}$. The key assumption behind incorporating this second term is to guide the model to focus more effectively on a specific condition of interest such as Alzheimer's Disease (AD).

Let $Data_{sample}$ be a randomly selected subset of training cells predicted as AD from the original dataset. It is used to fine-tune the new layers that constitute the second term and the secondary output, aiming to reproduce the predictions made by $Model_{SA}$ produces a secondary output, an (N+1)-dimensional vector denoted as $Output2_i$, which closely approximates the token relevance in accordance with external knowledge (ground-truth biological pathways encoded as vector). This process is formulated as follows:

$$\hat{P}_i, Output2_i = Transformer_{DSCGA}(E_i, C_i)$$
$$Output2_i = Agg(M_{A2})W_{agg} \tag{3}$$

Here, $Model_{DSCGA}(\cdot)$ refers to the fine-tuned model that incorporates the new layers and produces two intermediate processing outputs. $Agg(\cdot)$ is an aggregation function (sum) used to aggregate the $M_{A2}$ matrix for the condition of interest into a vector representing the importance of the different tokens (pathways). The key assumption behind this is that each score reflects the relevance of a biological pathway with respect to the predicted condition of interest. A high score indicates a more important token (pathway), while a low score corresponds to an irrelevant one. $W_{agg} \in \mathbb{R}^{(N+1)\times(N+1)}$ is a learnable weight matrix.

### 3.3.2 DYNAMIC SELECTION OF ATTENTION MATRICES STRATEGY

Using the trained $Model_{DSCGA}$ model, the next step requires a purposeful modification of the self-attention mechanism. This modification introduces a condition-driven mechanism that dynamically activates or deactivates the second attention matrix based on an input condition, allowing the model to adaptively adjust its attention computation in accordance with the context of the input tokens. This process is expressed as follows:

$$C_i = \text{Encode}(\arg\max(\text{Model}_{SA}(E_i)))$$

$$\text{Att}_{\text{matrix}} = \text{Softmax}\left(\frac{M_A + M_{A2} \cdot C_i}{\sqrt{d_k}}\right) \tag{4}$$

where $\arg\max(Model_{SA}(E_i))$ denotes the predicted label for the $i$-th input cell, produced by the $Model_{SA}$ based on standard self-attention. The function $\text{Encode}(\cdot)$ converts this label into a binary condition relative to the condition of interest. The main goal behind this strategy is to guide the model to uncover the relevant and irrelevant biological pathways associated with each condition of interest.

### 3.4 BIOLOGICAL INTERPRETATION-INFORMED ATTENTION SCORE

To enable token-level insights, it is important to examine the attention score matrix and aggregate the interactions of each token with others, resulting in an attribution score vector that highlights the meaningful pathways associated with each predicted disease-specific classification label. This process is formulated as follows:

$$Exp_i = Agg(softmax(S))$$
$$S = (Att_{matrix} \circ Repeat(Agg(Att_{matrix}))W_{agg}) \tag{5}$$

The core assumption behind $Exp_i \in \mathbb{R}^{(N+1)}$ is that the attention matrix captures the relationships between different tokens (pathways) and the cell itself. A high attention score indicates a more significant interaction, while a lower score suggests less relevance. Therefore, aggregating these interactions helps to identify the most relevant pathways corresponding to the predicted outcome label.

## 4 EXPERIMENTS

This section covers the experiments conducted to evaluate the effectiveness of our proposal. Section 4.1 presents the real-world datasets adopted in this work. Section 4.2 describes the methodology and evaluation measures. Section 4.5 summarizes the different state-of-the-art techniques. Finally, section 4.6 details the experimental results.

Note that detailed evaluation measures, implementation details, and hyperparameter settings with additional experiments are reported in the Appendix.

### 4.1 DATASETS

The experiments are carried out using two real-world single-cell datasets, called Seattle and ROSMAP. The detailed descriptions of the two datasets are as follows:

1. **Seattle dataset** (Gabitto et al., 2024): It is a public single-cell dataset, composed of millions of labeled cells. We sampled 50000 cells. Each one is labeled in one of two possible classes: AD and Normal (control). The cell labels were sampled in a balanced way. The distribution of labels is defined as follows: 25000 (50%) Normal labels, and 25000 (50%) AD labels.

2. **ROSMAP dataset** (Mathys et al., 2019): Religious Orders Study or the Rush Memory and Aging Project (ROSMAP) is a popular benchmark dataset. It consists of a set of instances. Each instance represents a single cell taken from a human brain donor. The different cells were classified into two separate categories: Normal, AD. The distribution of instance labels were as follows: 25000 (50%) Normal labels, 25000 (50%) AD labels. The total number of sampled cells adopted in this work is 50000. Each cell is associated with its corresponding label. The total number of genes used is 5000.

### 4.2 METHODOLOGY AND EVALUATION MEASURES

For the experimental study, each dataset is randomly partitioned into 2 parts, training and test sets. The test set is used for evaluating the model's performance, it consists of 20% of cells. While the training set composed of 80% of cells is adopted for training the models. This process is repeated five times, then the average is calculated.

We evaluated the model performance using the classification accuracy for cell disease classification. We assessed the biological interpretability using a top-k pathway metric, which measures the model's ability to identify ground-truth disease-relevant pathways. Full details of the evaluation measures are provided in Appendix B.1

### 4.3 BASELINES

In this study, we compare our proposed method against a set of baselines described below.

**Comparison-based attention mechanism:**

1. $Model_{Attentionsteering}$ (Zhang et al., 2024b): It modifies the self-attention mechanism by introducing attention steering, a technique designed to bias the attention weights toward domain-relevant tokens (e.g., biological pathways). It serves as a strong baseline for guided interpretability.

2. $Model_{SA}$ (Vaswani, 2017): This is a Transformer architecture based on the standard self-attention, where the attention scores are learned without any external guidance. It provides a reference point for unguided, vanilla attention mechanisms.

3. $CellEmbedding_{SA}$: It is derived from the $Model_{SA}$. This approach extracts token-level attention scores for each individual cell. These scores reflect how much attention each pathway receives per cell, enabling fine-grained interpretability.

4. Random Guess: A naive baseline assigns random attribution scores to pathways.

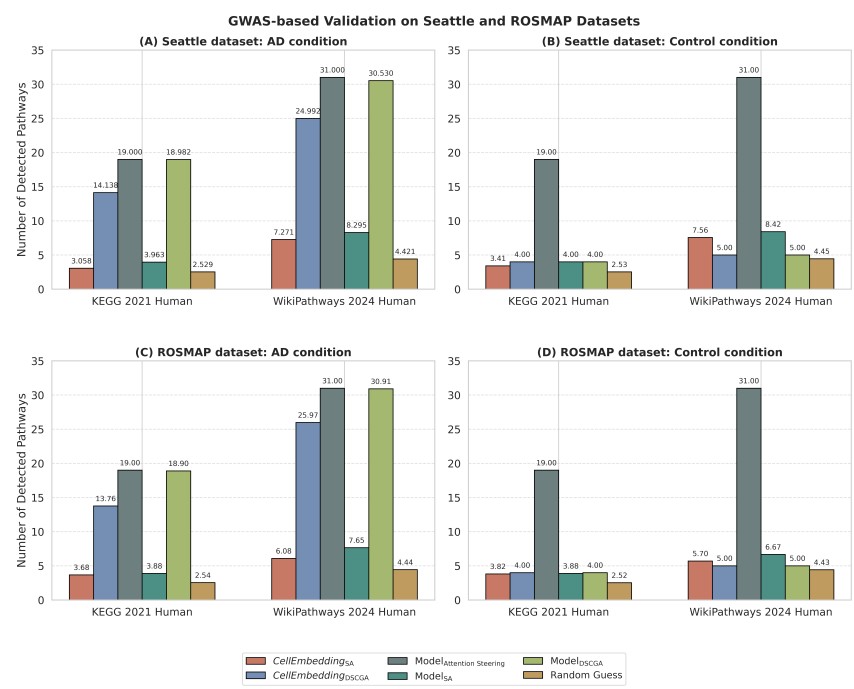

Figure 3: GWAS-based validation for two conditions AD and Control using Seattle and ROSMAP datasets.

5. $Model_{DSCGA}$: This is our interpretable transformer, which incorporates our DSCGA mechanism. It modifies the attention computation by regressing toward pathway relevance scores derived from GWAS priors.

6. $CellEmbedding_{DSCGA}$: Analogous to $CellEmbedding_{SA}$, this variant extracts per-cell attention scores from $Model_{DSCGA}$.

**Comparison of attribution-based XAI methods:**
We compared our technique with gradient-based attribution approaches, such as Integrated Gradient (Sundararajan et al., 2017) adopted in (Heimberg et al., 2025), DeepLIFT (Shrikumar et al., 2017), GradientSHAP (Scott et al., 2017), and Input×Gradient (Simonyan et al., 2013).

### 4.4 EXPERIMENTAL RESULTS

This section covers the evaluation protocol defined to prove the effectiveness of our proposal in terms of biological interpretability.

**GWAS-based validation:** This step is intended to uncover the relevant pathways that contribute to the predicted condition (label). Figure 3 presents the results on the Seattle and ROSMAP datasets. Specifically, it depicts the number of correctly detected AD-related pathways achieved by various baselines compared to our proposed model based DSCGA mechanism. The results reveal that our proposed architecture substantially outperforms the other baselines on the two datasets. In particular, our proposal is able to yield a much superior number of correctly detected pathways than the other competitors for AD condition, while for Control condition, the number of pathways pinpointed is low which proves that our model distinguishes between the pathways relevant to each condition separately. However, the Attention steering is biased toward detecting AD pathways correctly but fails for the control. This fact gives a clear indication that our proposal can enhance the performance of other existing architectures.

**Validation of attribution-based methods explanations:** To showcase the performance of our DSCGA, we employed four popular XAI methods. Each method assigns scores to

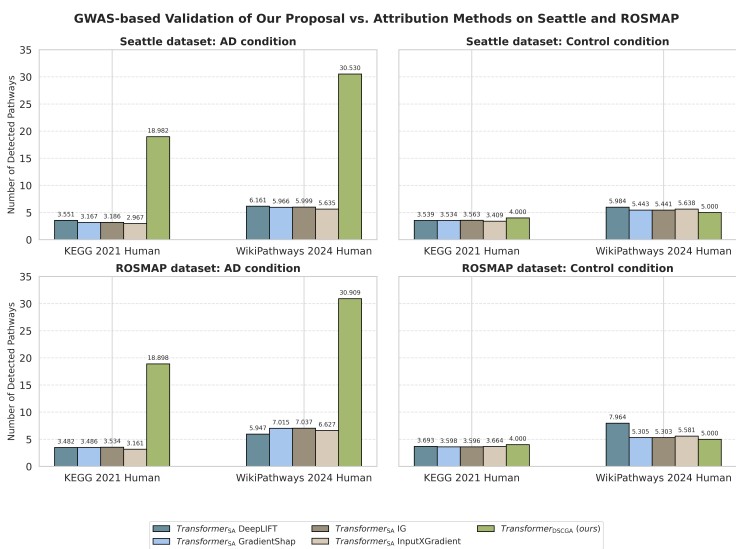

Figure 4: Comparison of our proposal vs. attribution-based methods explanations for AD and Control conditions using Seattle and ROSMAP datasets.

pinpoint the importance of the tokens (i.e., pathways) under specific pathological conditions. Figure 4 illustrates the number of pathways detected using the Seattle and ROSMAP datasets. These results clearly demonstrate that the explanations provided by XAI methods are not consistent with biological insights (low values for the two conditions). In contrast, our proposal offers a solution that enables the development of biologically interpretable transformers (high values for AD condition and low for Control).

**Cell disease classification:** We evaluated the model performance using Seattle and ROSMAP. The results show that our proposed DSCGA approach improves the biological interpretability with minimal loss in classification accuracy compared to standard self-attention and attention steering baselines. Detailed results and comparisons are provided in Appendix A.3.

**Impact of dynamic selection attention matrices strategy:** We assessed the impact of dynamic selection strategy by comparing DSCGA performance with and without it. The results proved that the dynamic selection improves the model's ability to identify pathways relevant to the predicted condition. Details results are presented in Appendix A.4.

**Supplementary experiments using a new larger dataset:** We adopted a third dataset and performed comparisons of cell-level versus donor-wise splits, biological interpretability validation, top-k pathway overlap, and statistical tests. Results confirm the effectiveness of our proposal. The details are provided in the Appendix B.4.

## 5 Conclusion

In the present work, we introduced an interpretable Transformer-based architecture that incorporates a dynamic condition guided attention mechanism, enabling the model to produce biologically meaningful interpretations aligned with established neuroscience knowledge. Our evaluation on two benchmark single-cell transcriptomics datasets demonstrates that the proposed approach outperforms existing baselines in terms of biological interpretability. Our collective results highlight the value of integrating domain-specific knowledge into the design of transformer learning systems to improve interpretability in biomedical applications. As future work, we plan to explore advanced mechanisms to mitigate attention head polysemanticity, fostering specialization of attention heads toward distinct roles.

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

## APPENDIX

This supplementary material provides additional details regarding the data preprocessing and input representation, model components and architecture, hyper-parameter settings and additional experiments using another larger seattle dataset.

## A    DATA PREPROCESSING AND INPUT REPRESENTATION

This section describes the data preprocessing, specifically, the gene set enrichment analysis, and token definition including, encoding pathways and cell representation.

Given a dataset $D_{cells} = \{(Cell_1, L_1), (Cell_2, L_2), \ldots, (Cell_{|D_{cells}|}, L_{|D_{cells}|})\}$, which is composed of $|D_{cells}|$ instances. Each instance $(Cell_i, L_i)$ represents a single cell denoted $Cell_i$ and its corresponding label $L_i$ indicating either AD or healthy condition. Let $C = \{Cell_1, Cell_2, \ldots, Cell_{|D_{cells}|}\}$ and $G = \{G_1, G_2, \ldots, G_Z\}$ be the sets of cells and genes, respectively. $Z = |G|$ stands for the total number of genes. Each cell $Cell_i = (G_1, G_2, \ldots, G_Z)$ is made up of $Z$ gene count measurements. Each gene $G_j \in \mathbb{R}$ has an expression value indicating the level of activity within the cell.

### A.1    GENE SET ENRICHMENT ANALYSIS

This step is tailored to group the set of highly variable genes, denoted $G$, into gene sets corresponding to known biological pathways. This process can be expressed as follows:

$$\mathcal{P} = \{\mathcal{P}_i, i = 1, \ldots, |\mathcal{P}|\} = \text{GSEA}(G) \tag{6}$$

The symbol $\mathcal{P}$ represents the set of pathways, where each one corresponds to a gene set. The function $GSEA(.)$ performs gene set enrichment analysis and returns the relevant pathways selected based on their *p-values* $\leq 0.01$. The term $|\mathcal{P}|$ refers to the total number of pathways.

Given the set of pathways $\mathcal{P}$ and genes $G$, we define a sparse matrix, denoted $SM \in \mathbb{R}^{|\mathcal{P}| \times Z}$, where each row corresponds to a pathway and the columns stand for the set of genes. Each element of $SM$ is computed as follows:

$$SM_{i,j} = \begin{cases} 1 & if \quad value(G_j) \neq 0 \quad \& \quad G_j \in \mathcal{P}_i \\ 0 & otherwise \end{cases} \quad (7)$$

where $value(.)$ is a function that returns the gene expression value.

## A.2 Token definition: Encoding pathways and cell representation

Given the set of curated biological pathways $\mathcal{P}$. The i-th pathway $P_i$ can be encoded as a $Z$-dimensional vector. Let $IP_i \in \mathbb{R}^Z$ be an indicator vector representing the pathway $P_i$. For example, suppose the total number of highly variable genes equals 10, and $P_1 = \{G_1, G5, G8, G_9\}$ is composed of four genes. The elements of $IP_1 \in \mathbb{R}^{10}$ corresponding to the gene indices (1, 5, 8, 9) are set to 1, while all other elements are set to 0. This vector encodes the presence of genes associated with pathway $P_1$.

Consequently, the n-th cell $Cell_n \in \mathbb{R}^Z$ can be defined as a sequence of $|\mathcal{P}|$ pathways, where each pathway is encoded as the element-wise product of the indicator vector and the gene expression values of the cell. This can be expressed as follows:

$$Input_n = pathways\_vect_n = \{T_1 = IP_1 * Cell_n, T_2 = IP_2 * Cell_n, \ldots, T_{|\mathcal{P}|} = IP_{|\mathcal{P}|} * Cell_n\}$$
$$pathways\_vect_n = \{T_1, T_2, \ldots, T_{|\mathcal{P}|}\}$$
$$(8)$$

Here, $Input_n$ and $pathways\_vect_n$ refer to the n-th input cell, represented as set of $|\mathcal{P}|$ tokens $(T_1, T_2, \ldots, T_{|\mathcal{P}|})$, where each token is equivalent to a biological pathway.

In this work, we used an additional token, denoted $T_{|\mathcal{P}|+1}$, representing the cell itself (all genes expression). Consequently, the input comprises $|\mathcal{P}| + 1$ tokens corresponding to $|\mathcal{P}|$ pathways and the cell, defined as follows:

$$Input_n = \{T_1 = IP_1 * Cell_n, T_2 = IP_2 * Cell_n, \ldots, T_{|\mathcal{P}|} = IP_{|\mathcal{P}|} * Cell_n, T_{|\mathcal{P}|+1} = Cell_n\}$$
$$Input_n = \{pathways\_vect_n, T_{|\mathcal{P}|+1} = Cell_n\}$$
$$pathways\_vect_n = \{T_1, T_2, \ldots, T_{|\mathcal{P}|}\}$$
$$(9)$$

## A.3 Cell disease classification

For the cell disease classification task, the experiments are performed using two real-world single-cell datasets and two popular biological pathways databases, namely KEGG 2021 Human, and WikiPathways 2024 Human. Each experiment is repeated five times. The main objective is to evaluate the performance of the transformer based on the standard self-attention, and compare it with our proposed DSCGA and attention steering baseline.

Figure 5 depicts the results obtained using two datasets. It presents comparative results in terms of classification accuracy, demonstrating the efficiency of the proposed approach against other baselines. It can be observed that our approach achieves better results than the model based on steering attention. The original model based on standard self-attention outperforms each of the other architectures. This was expected as we need to consider the trade-off between accuracy drop and biological Interpretability.

## A.4 Impact of dynamic selection attention matrices strategy

To evaluate the impact of dynamically selecting attention matrices strategy, we compared the biological interpretability of attention scores of DSCGA with dynamic selection strategy vs. DSCGA without it, in terms of their ability to uncover the relevant pathways aligned with the predicted condition. Figures 6 and 7 depict the results using Seattle and ROSMAP datasets, respectively. Tables 1 and 2 present the detailed number of detected AD-related pathways.

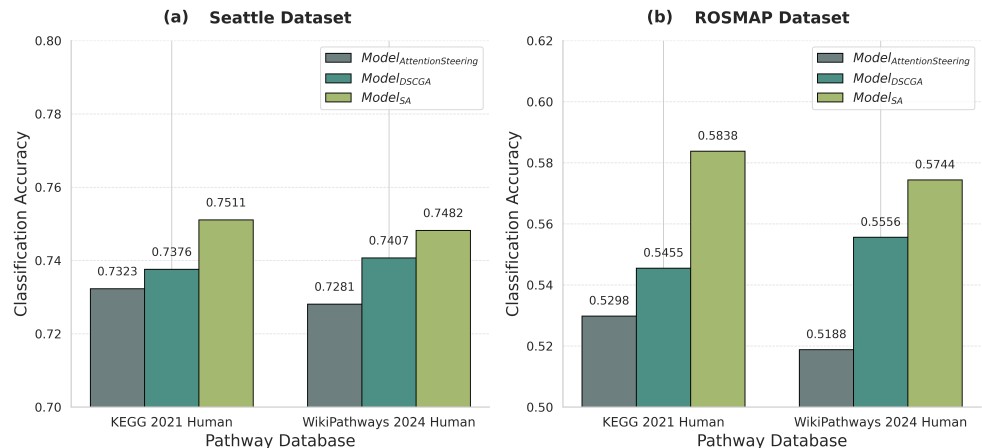

Figure 5: Classification performance comparison using Seattle and ROSMAP datasets.

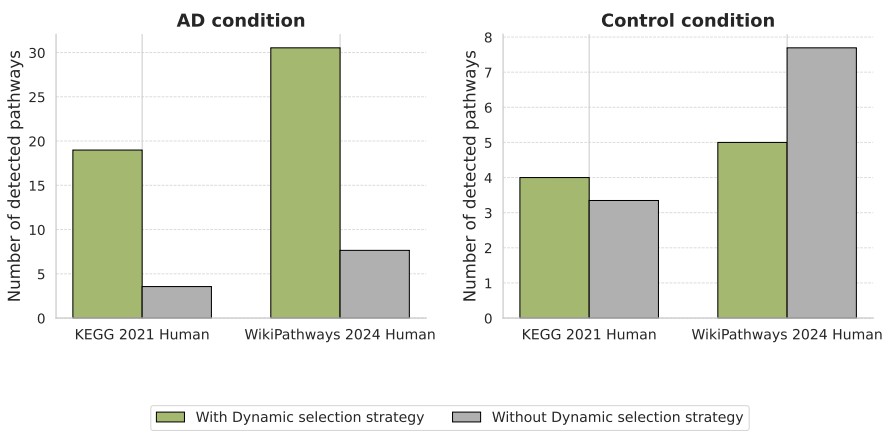

Figure 6: Impact of using the dynamic selection of attention matrices strategy (Seattle dataset)

Table 1: Impact of using the dynamic combination strategy seattle dataset

| Method | KEGG 2021 Human | | WikiPathways 2024 Human | |
|---|---|---|---|---|
| | AD | Control | AD | Control |
| $Model_{DSCGA}$ | $18.9816 \pm 0.0312$ | $4.0000 \pm 0.0000$ | $30.5300 \pm 0.6314$ | $5.0000 \pm 0.0000$ |
| $Model_{DSCGA}$ without dynamic combination | $3.5582 \pm 0.4010$ | $3.3469 \pm 0.7945$ | $7.6483 \pm 1.8629$ | $7.6931 \pm 1.8334$ |

Table 2: Impact of using the dynamic combination strategy ROSMAP dataset

| Method | KEGG 2021 Human | | WikiPathways 2024 Human | |
|---|---|---|---|---|
| | AD | Control | AD | Control |
| $Model_{DSCGA}$ with dynamic combination strategy | $18.8976 \pm 0.0660$ | $4.0000 \pm 0.0000$ | $30.9089 \pm 0.1025$ | $5.0000 \pm 0.0000$ |
| $Model_{DSCGA}$ without dynamic combination strategy | $3.5460 \pm 0.3740$ | $3.3402 \pm 0.2512$ | $7.0880 \pm 0.6894$ | $6.4851 \pm 0.5669$ |

For the AD condition, a higher value indicates better detection and more accurate identification of AD-related pathways, whereas for the control condition, a lower value reflects better performance, as most AD-related pathways are not highlighted. Our findings show that the

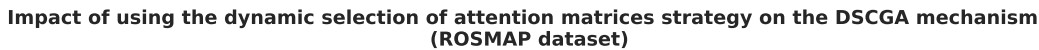

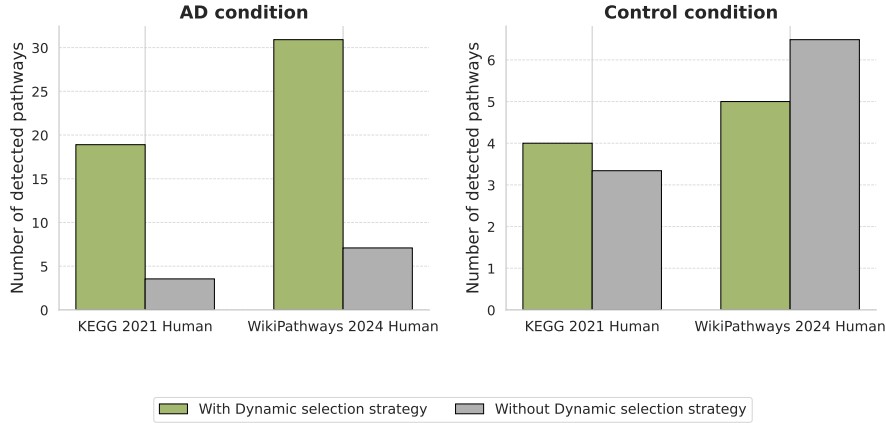

Figure 7: Impact of using the dynamic selection of attention matrices strategy (ROSMAP dataset)

dynamic combination of attention matrices strategy enhances biological interpretability and improves the quality of pathways-level insights.

## B  SUPPLEMENTARY DETAILS AND EXPERIMENTS

This section covers the evaluation measures, the optimal hyperparameters used in our experiments, along with implementation details and additional experimental results using a new larger dataset.

### B.1  EVALUATION MEASURES

In this work, we assessed the performance of several models for cell disease classification and biological interpretability tasks.

For the cell disease classification, we employed the classification accuracy (Lin et al., 2021; Arnab et al., 2021; An et al., 2021; Dosovitskiy et al., 2021; Liu et al., 2021). In general, a higher value indicates better performance. Meanwhile, for the biological interpretability task, we defined our own protocol to assess the learning ability of the model to capture the biological pathways, quantify their importance, and generate biological interpretations associated with the different predicted conditions. To do so, we employed the following metrics:

**Top-k pathways ratio:** Let $k$ denote the number of ground-truth AD-related biological pathways. For the i-th cell, let $Exp_i$ be the interpretation derived from attention scores, where each element represents a score assigned to a biological pathway. We extract the top-$k$ elements from $Exp_i$, denoted as $Exp_i^{AD}$ for AD-predicted condition and $Exp_i^{Control}$ for control-predicted condition. These are compared against a set of $k$ ground-truth AD-related pathways that include at least one gene identified in Genome-Wide Association Studies (GWAS). This process is repeated for all cells in the test set. Formally, let $G_{pathways}$ be the set of $k$ ground-truth AD-related pathways, and let top-$k(.)$ be a function that returns $k$ relevant pathways for the $i$-th cell. The number of correctly identified pathways is calculated as follows:

$$R_{AD} = \frac{\sum_{i=0}^{M_{AD}} |G_{pathways} \cap \text{top-}k(Exp_i^{AD})|}{M_{AD}}$$

$$R_{Control} = \frac{\sum_{i=0}^{M_{Control}} |G_{pathways} \cap \text{top-}k(Exp_i^{Control})|}{M_{Control}}$$

(10)

where $M_{AD}$ and $M_{Control}$ refer to the number of cells predicted as AD and control, respectively. Specifically, a higher $R_{AD}$ value indicates that more AD-related pathways were correctly detected. Conversely, for $R_{Control}$, a smaller value reflects better performance, as it indicates that pathways not related to the AD condition were detected.

## B.2 Hyper-parameter settings

In this work, several experiments were carried out to select the optimal hyper-parameters. The total number of highly variable genes was fixed at 5000, which defines the size of each input token. We adopted raw count matrix. The embedding size for both pathways and cells tokens was set to 128. Each dense layer consisted of a number of neurons equals the number of input tokens. The number of epochs for pre-training the model was set to 5. The batch size equals 256. The parameters $D$ and $D_k$ were set to 128 and 32, respectively. The learning rate is fixed at $1 \times 10^{-3}$. We conducted several experiments by using two pathway databases, namely KEGG 2021 Human and WikiPathways 2024 Human. We varied the number of transformer blocks and attention heads. Since interpretation in our framework involves aggregating attention scores across heads and tokens, we observed that increasing the number of heads did not significantly alter interpretability results. Additionally, to avoid overfitting, we used the optimal number of heads. Similarly, increasing the number of transformer blocks did not yield improved performance in terms of discriminative capacity. Due to the absence of pre-trained models tailored to our specific task, the model was trained from scratch, and we set the number of block at one. Post-training was conducted using a sample of 20,000 cells. As the model produces two outputs, we employed cross-entropy loss for classification and mean squared error (MSE) for the regression objective.

## B.3 Implementation details

In this paper, the experiments were conducted on a cluster. We used 900 GB RAM, and GPU Nvidia RTX 8000. All the deep learning techniques for cell disease classification were written in Python. The details of the operating system and the different libraries are: Python 3.8, Pytorch 1.11.0, Captum, Scanpy, GSEApy, Linux.

## B.4 Additional results on biological interpretability with an extra dataset

We carried out additional experiments using a third dataset (named larger seattle dataset) consisting of nearly 84000 cells of 89 donors. The choice is mainly based on the availability of ground-truth annotations (domain-specific external knowledge) required to validate the interpretability of our approach. To demonstrate its effectiveness, we considered two scenarios, called (1) donor-wise splits, and (2) cell-level splits. (1) Donor-wise splits: we split the third dataset (large Seattle) into 80% of donors in training, the remaining 20% donors for testing the model. The training and test data are balanced. (2) Cell level splits: this consists in partitioning the training and test sets at the cell level, i.e., we randomly divide the 84000 cells into training and test sets. The training and test data are balanced.

### B.4.1 Classification comparison of cell-level splits vs. Donor-wise splits

The first goal is to compare the predictive performance using different data splits (cells vs donors). We used the large Seattle dataset. Figures 8-9 depict the results using KEGG 2021 and WikiPathways 2024 Human, respectively. The interpretable model is based on our proposed DSCGA attention mechanism, while the original model relies on vanilla attention.

The results indicate that both the original model and the interpretable model demonstrate nearly identical predictive performance with minimal differences. When we train the models using cell-level splits, higher accuracy is achieved after several epochs. In contrast, the training with donor-wise splits reaches the optimal performance in just one epoch.

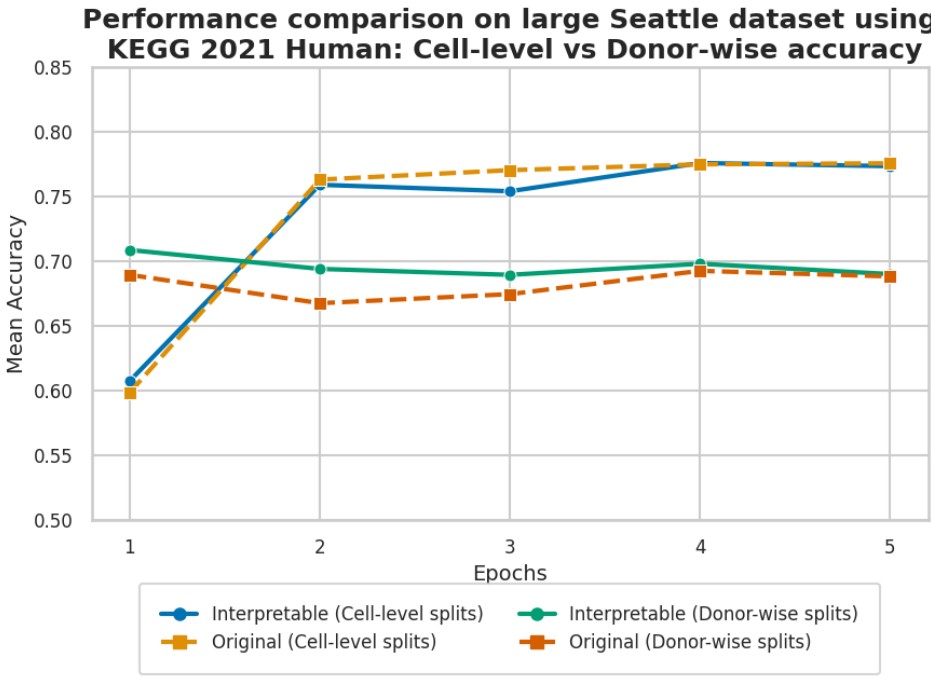

Figure 8: Classification comparison of cell-level splits vs. donor-wise splits using a third dataset set (large seattle) and KEGG 2021 pathways database.

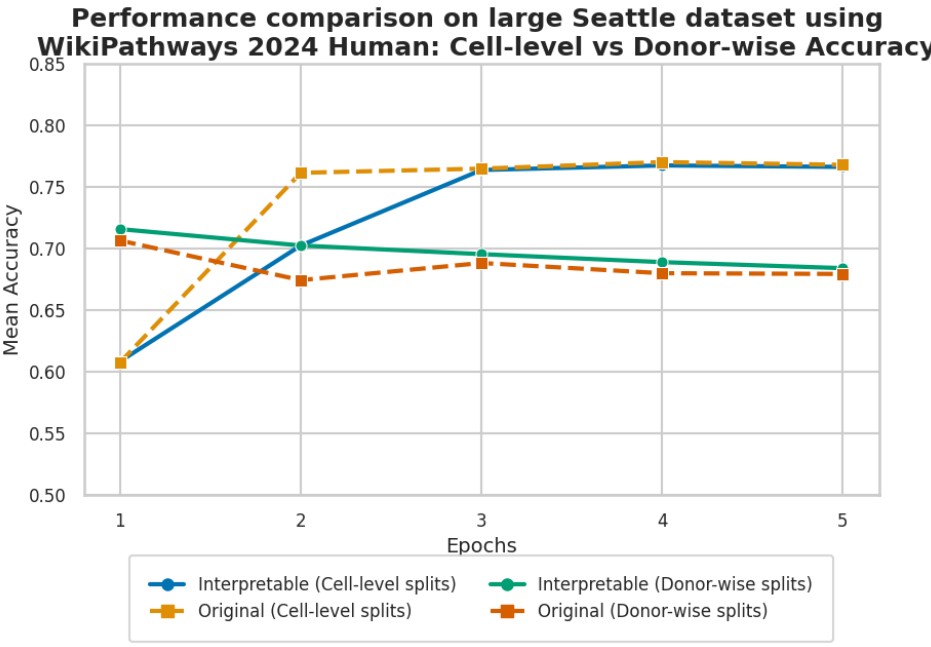

Figure 9: Classification comparison of cell-level splits vs. donor-wise splits using a third dataset set (large seattle) and Wikipathways 2024 Human database.

Figure 10: Interpretability validation using a large Seattle dataset and donor-wise splits.

### B.4.2 BIOLOGICAL INTERPRETABILITY VALIDATION USING CELL-LEVEL SPLITS VS. DONOR-WISE SPLITS

This section aims to assess the biological validity of interpretations using different data splits. Figures 10-11 display the results based on donor-wise splits and cell-level splits, respectively. Each one details the number of detected pathways based on KEGG 2021 and WikiPathways 2024 Human for AD and Control conditions.

Our proposed DSCGA attention demonstrates a significantly higher number of correctly identified pathways for the Alzheimer's disease (AD) condition compared to random guessing and vanilla attention baselines. In contrast, the number of pathways pinpointed for the control condition is low, indicating that our model effectively distinguishes between pathways that are relevant to each condition separately. In addition, when we compared the interpretability of DSCGA attention against our DSCGA attention without the dynamic activation gating. It turns out that the dynamic activation gating contributes significantly to enhancing the quality of class-specific interpretations.

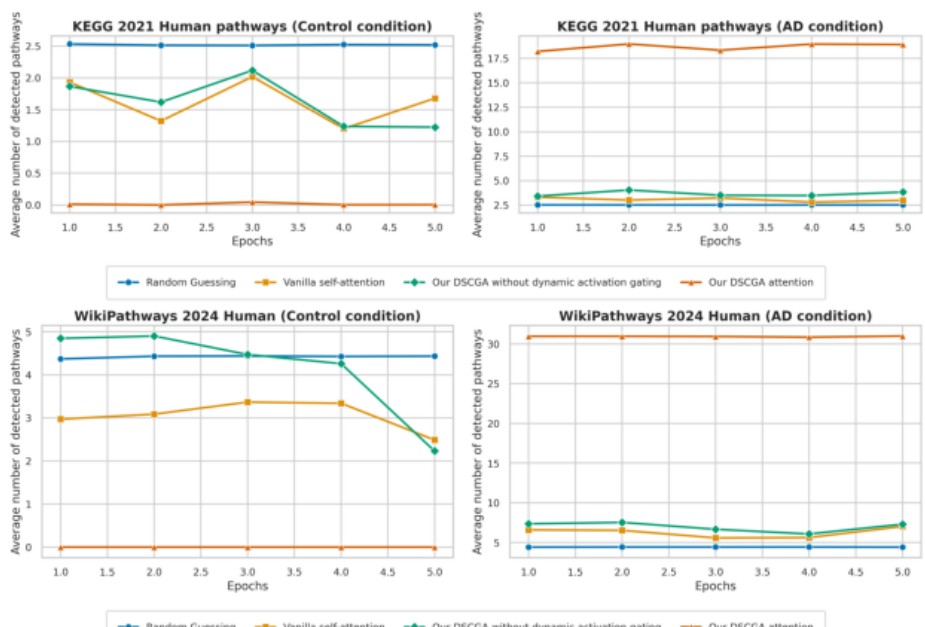

Figure 11: Interpretability validation using a large Seattle dataset and cell-level splits.

Figure 12: Top-k pathway overlap comparison between control and AD across donor-wise splits using the KEGG database and the large Seattle dataset

### B.4.3 Top-k pathway overlap comparison

The next objective is to evaluate the quality of biological interpretability by studying the impact of varying the number of top-k most relevant biological pathways (tokens). We evaluated three different values for the parameter k (i.e., 5, 10, 15). Figures 12-13 present the results based on donor-wise splits and cell-level splits using KEGG database pathways and a large Seattle dataset. Figures 14-15 illustrate the results based on donor-wise splits and cell-level splits using the WikiPathways 2024 Human database.

The results support the previous conclusions that the biological interpretability of our approach remains consistent regardless of the parameter $k$ or data split.

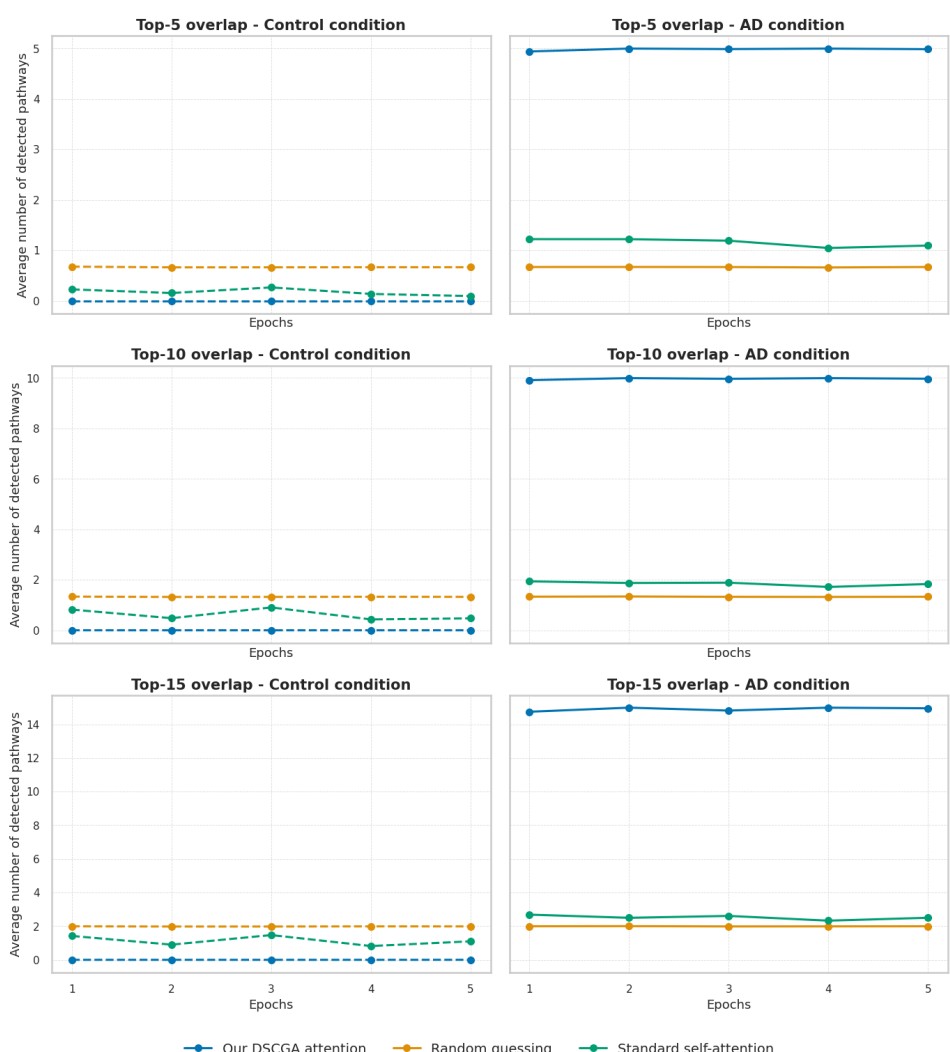

Figure 13: Top-k pathway overlap comparison between control and AD across cell-level splits using the KEGG database and the large Seattle dataset

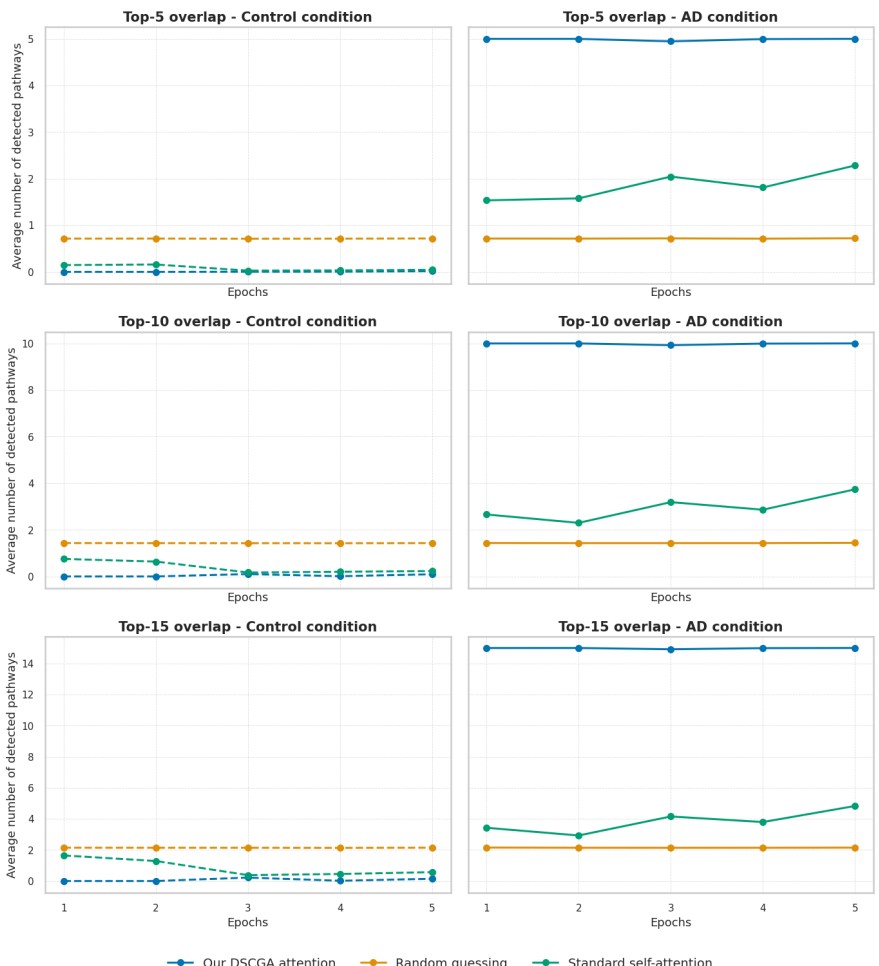

Figure 14: Top-k pathway overlap comparison between control and AD across donor-wise splits using the WikiPathways database and the large Seattle dataset

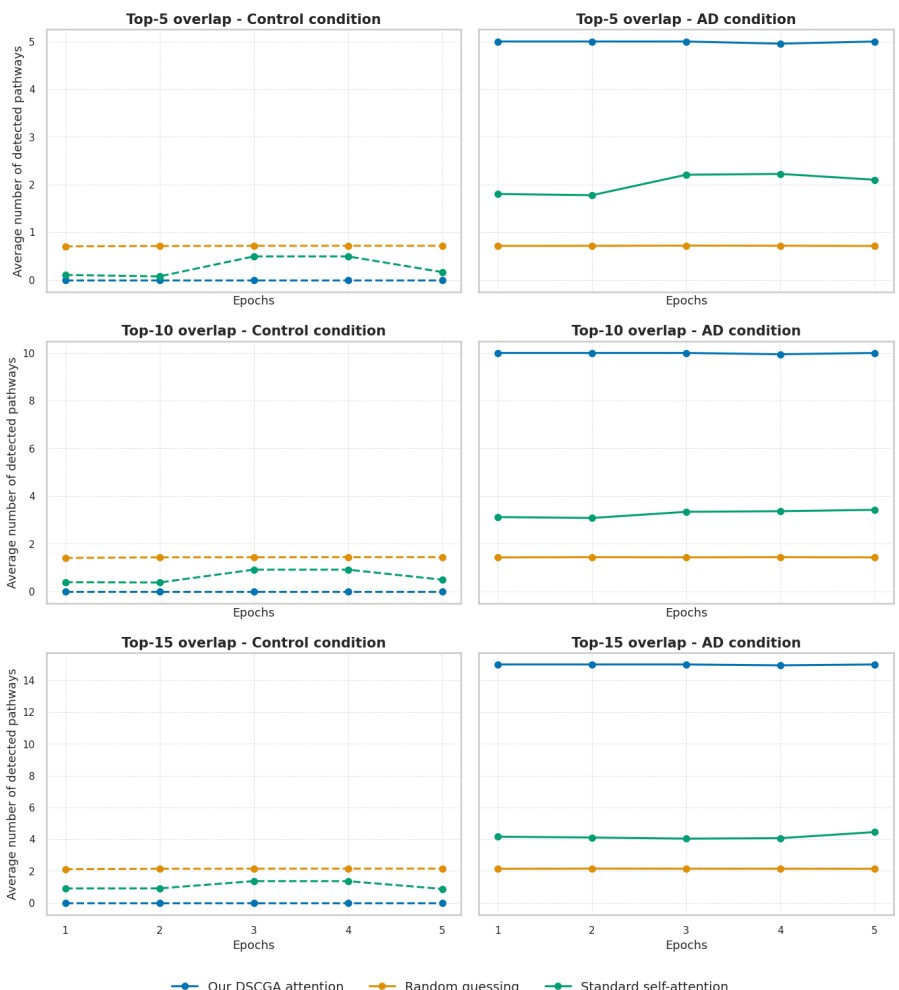

Figure 15: Top-k pathway overlap comparison between control and AD across cell-level splits using the WikiPathways database and the large Seattle dataset

### B.4.4 STATISTICAL TEST

We employed a paired t-test to compare the predictive performance difference between the original model based on vanilla self-attention and the interpretable model based on our DSCGA attention. We considered the two types of splits, cell-level splits and donor-wise splits. Table 1 shows the results based on KEGG and Wikipathways databases.

Table 3: Statistical comparison of classification methods (Interpretable model based on our DSCGA vs. original model based on standard self-attention) using paired t-tests on the large Seattle dataset.

| Pathway Database | Split | t-statistic | p-value |
|---|---|---|---|
| Seattle and KEGG 2021 Human | Donor-wise | 5.2746 | 0.0062 |
| | Cell-level | -1.1879 | 0.3006 |
| Seattle and WikiPathways 2024 | Donor-wise | 1.9082 | 0.1290 |
| | Cell-level | -1.1998 | 0.2964 |

All the p-values except for one were higher than 0.05, which indicates there is no statistical significance between them. This demonstrates that the predictive performance of both models is quite similar in most cases, regardless of the data splits used.

## C  MODEL COMPONENTS AND ARCHITECTURE

This section covers the core components of the architecture, including the self-attention mechanism, transformer block, and the transformer architecture based on the standard self-attention mechanism.

### C.1  CORE COMPONENTS

#### C.1.1  SELF-ATTENTION MECHANISM

The self-attention mechanism underpins much of modern transformer-based large language models (LLMs). Given a set of input tokens $(T_1, \ldots, T_{(|\mathcal{P}|+1)})$, each one is embedded into a vector space. We denote the corresponding embeddings as $E = (E_1, \ldots, E_{(|\mathcal{P}|+1)})$. The goal is to map $E$ into a different embedding space that captures the semantic relationships between the pathways and the cell, resulting in transformed set of representations $Y = (Y_1, \ldots, Y_{|\mathcal{P}|+1)|})$. Each output vector $Y_j$ depends on all the tokens. This process can be written as follows:

$$Y = \text{Attention}(E) = \text{Softmax}\left(\frac{M_A}{\sqrt{d_k}}\right) V$$

$$M_A = QK^T$$

$$Q = EW^q; \ K = EW^k; \ V = EW^v$$

(11)

where the function $Attention(.)$ denotes the standard self attention. $W^q \in \mathbb{R}^{D \times D_q}$, $W^k \in \mathbb{R}^{D \times D_k}$, $W^v \in \mathbb{R}^{D \times D_v}$ are weight matrices used to linearly transform the input tokens using fully connected layers with linear activation. $D_k$, $D_v$, and $D_q$ represent the dimensions of the key, value, and query vectors, respectively. $D_q = D_k$ and $D_v = D$ ensures the output representation $Y$ matches the input dimensionality. $M_A \in \mathbb{R}^{(|\mathcal{P}|+1)|) \times (|\mathcal{P}|+1))}$ denotes the attention score matrix, where each element represents the interaction between a pair of tokens. Notably, the last row of $M_A$ captures the interactions between all pathways and the entire cell context.

#### C.1.2  TRANSFORMER BLOCK

Let $block(.)$ be a function representing a Transformer block adopted in this work. It consists of the following layers:

- **Input Layer**: This layer takes each cell as a sequence of $|\mathcal{P}|+1$ embedding vectors $E$ representing the pathways and the cell.

- **Attention Layer**: This is the self-attention that refines the contextual representation.

- **Residual connections**: It implements dropout and adds the original input to the output of the attention layer.

- **Post-Normalization Layer**: This applies normalization to the input embeddings.

- **FFN**: This is a feed-forward network (FFN), two-layer fully connected layers with the SeLU (Scaled Exponential Linear Unit) activation function.

## C.2 TRANSFORMER-BASED SELF-ATTENTION ARCHITECTURE

The complete Transformer-based standard self-attention used to classify the i-th cell $Input_i$ is written as follows:

$$\hat{P}_i = Model(Input_i) \tag{12}$$

The model's architecture denoted $Model(.)$ is based on the standard self-attention. It is made up of the following layers:

- **Embedding Layer**: This layer takes each cell $Input_i$, a sequence of sparse pathways representation and convert it into a sequence of $(|\mathcal{P}|+1)$ embedding vectors $E_i$ representing the i-th cell $Cell_i$. It employs a dense layer with ReLu (Rectified Linear Unit) activation function. This layer is expressed as follows:

$$E_i = Embedding\_layer(Input_i) = Embedding\_layer(concat(pathways\_vect_i, Cell_i)) \tag{13}$$

  where $E_i \in \mathbb{R}^{(|\mathcal{P}|+1) \times D}$ refers to the embedded representation of the entire cell $Cell_i$. $Embedding\_layer(.)$ represents the embedding layer function.

- **Transformer Block**: This block is defined in the previous section. It can be written as follows:

$$O_{block}^i = block(E_i) \tag{14}$$

  Here $O_{block}^i \in \mathbb{R}^{(|\mathcal{P}|+1) \times D}$ is the block's output with $D$ is the size embedding.

- **Convolution1D Layer**: This 1D convolutional layer that transforms each cell's embedding vectors into $(|\mathcal{P}|+1)$-dimensional vector. Each element corresponds to an aggregated feature for a particular pathway, with the last one representing the complete set of the cell's genes. It is expressed as follows:

$$O_{Conv}^i = Conv1D(O_{block}^i) \tag{15}$$

  Where $O_{Conv}^i \in \mathbb{R}^{(|\mathcal{P}|+1)}$ is the convolution layer output. The i-th element corresponds to the i-th biological pathway.

- **Dense layer**: It is a fully connected layer of $(|\mathcal{P}|+1)$ nodes with SeLU as nonlinear activation function. It is written as follows:

$$O_{Dense}^i = SeLu(Dense(O_{Conv}^i)) \tag{16}$$

  where $Dense(.)$ is a function representing a dense layer. $O_{Dense}^i \in \mathbb{R}^{(|\mathcal{P}|+1)}$ is a vector.

- **Softmax Layer**: This is a fully-connected layer with softmax activation function. It estimates the probability distribution of different conditions. It is defined as follows:

$$\hat{P}_i = Softmax(O_{Dense}^i) \tag{17}$$

  where $\hat{P}_i \in \mathbb{R}^2$ is a probability distribution over the different conditions (AD and Control). $Softmax(.)$ refers to the output layer. $O_{Dense}^i \in \mathbb{R}^{(|\mathcal{P}|+1)}$ is a vector.

