# OpenReview forum: "Interpretable Transformers by Condition Guided Self-Attention"
_ICLR.cc/2026/Conference — Submitted to ICLR 2026_

### Official Review · Reviewer_LEho · 2025-10-29

**Soundness:** 2
**Presentation:** 2
**Contribution:** 2
**Rating:** 2
**Confidence:** 3

**Summary:**

Distinguishing Alzheimer’s disease (AD) from healthy brain cells using high-dimensional single-cell transcriptomic data is a challenging classification task. To improve performance and explainability simultaneously, the paper introduces an architecture with a novel attention mechanism, Disease-Specific Conditional Guided Self-Attention (DSCGA). The main idea is to tailor the attention pattern to include domain-specific modeling. The approach maps gene expression to a dictionary of known biological pathways; the proposed attention mechanism then amplifies attention to pathways relevant to the specific condition. This attention score is progressively merged with the standard self-attention score and refined during post-training. The method is evaluated on the Seattle and ROSMAP datasets and appears to learn to focus on AD-related pathways.

**Strengths:**

- The research setting is highly relevant.
- The idea to incorporate expert knowledge given in a format that is hard to formulate in a language format into an existing model is novel and interesting.

**Weaknesses:**

- The individual components are not sufficiently motivated. It is unclear why these design choices were made and what specific purposes they serve. An ablation study would support these choices, including the two training stages.

- The comparison across approaches is limited. The proposed method has access to significantly more information, so it is unsurprising that it performs better. A simple baseline would be to train a standard adapter for the extra tokens E_1 ... E_N.

- The more interpretable model comes at the cost of lower accuracy. This raises the question: is a method that sacrifices accuracy for better attribution to “important” pathways more trustworthy? In other words, if attending to these pathways leads to worse accuracy, are they truly important, and is this the right evaluation scheme? In particular, GWAS often identifies singular pathway correlations, while combinations of multiple pathways may be more predictive; something a properly trained attention mechanism could exploit.

- The evaluation could benefit from ablating the “important” pathways to verify that they truly influence the model’s predictions.

- The attribution methods you use in Figure 4 were mostly developed for non-attention architectures. Hence, methods that are specialized for transformers would greatly improve the evaluation[1].


Minor:
- The template appears to be altered.
- Missing ".": "observation specifically" line 26
The related work section is very long. You could save quite some space here and focus more on results and method.
- The type setting of the equations could benefit from using \text in math environments.

[1] Walker, C., Ahmed, M. R., Jha, S. K., & Ewetz, R. (2025, April). Explaining ViTs Using Information Flow. In International Conference on Artificial Intelligence and Statistics (pp. 2440-2448). PMLR.

**Questions:**

- What is the motivation to use an extra attention mechanism, instead of using an adapter to project the tokens into the LLM token-space?
- Why is this second fine-tuning stage necessary?
- Your evaluation metric for explainability depends on a hyperparameter k. How do the results change with varying k?
- What is the domain of the condition scalar, and how is it determined?
- Did you further investigate which pathways the standard transformer used and if they are biologically relevant?

---

### Official Review · Reviewer_1BwN · 2025-10-30

**Soundness:** 3
**Presentation:** 2
**Contribution:** 2
**Rating:** 4
**Confidence:** 3

**Summary:**

This paper introduces Disease-Specific Condition Guided Self-Attention (DSCGA), an interpretable Transformer mechanism for classifying Alzheimer’s disease (AD) cells from single-cell transcriptomic data. Instead of relying on post-hoc explanation, DSCGA integrates external biological knowledge (pathway-level priors) directly into the attention computation through a condition-guided term that dynamically focuses on disease-relevant pathways. The method is evaluated on two Alzheimer’s datasets (Seattle and ROSMAP), showing that DSCGA substantially increases the number of correctly detected AD-related pathways (e.g., from 3.96 to 18.98 on KEGG) while maintaining classification accuracy comparable to standard Transformers.

**Strengths:**

The paper introduces a principled and biologically grounded approach to interpretability by embedding condition-guided attention directly within the Transformer architecture. Unlike post-hoc explanation methods, DSCGA achieves intrinsic interpretability—its attention weights inherently reflect disease-relevant biological pathways through a condition-aware modulation term informed by external pathway knowledge (e.g., KEGG, WikiPathways). This design allows the model to dynamically focus on mechanistically meaningful gene sets under different disease contexts, providing interpretable attention maps without sacrificing predictive performance. The idea is conceptually elegant, empirically validated, and potentially generalizable to other biomedical domains where explainability is essential.

**Weaknesses:**

- Disease-specific pathway data is relatively scarce and incomplete, which may limit the model's generalizability.
- Evaluations are limited to AD-related datasets and related pathway databases. It is unclear whether DSCGA generalizes to other diseases, tissues, or data modalities.
- As interpretability is central to this work, the authors should provide case studies demonstrating the biological significance of the explanations generated by the model, thereby further validating its effectiveness.
- Figure 1 does not appear to contain much information. It is recommended to compare the differences between DSCGA and SA using a format similar to Figure 2. By correlating with specific formulas, this approach can help readers better understand the method.

**Questions:**

- How robust are the interpretability and classification gains to errors or incompleteness in the pathway databases?
- Have you performed perturbation-based tests (e.g., ablate high-attention genes or pathways and measure downstream prediction drop) to quantify whether identified attention weights are causally important for predictions?
- The primary interpretability assessment measures the number of detected pathways. While this reflects overlap, it does not quantify pathway importance ranking or enrichment significance. Are there more granular evaluation methods available?

---

### Official Review · Reviewer_9LWb · 2025-10-30

**Soundness:** 2
**Presentation:** 2
**Contribution:** 2
**Rating:** 2
**Confidence:** 4

**Summary:**

This paper considers the task of interpretable predictions of labeling single cell whether they show signs of Alzheimer’s Disease. The authors suggest a mechanism to steer a transformer to learn about pathways relevant for the disease by an attention mechanism they term Disease-specific Conditional Guided Self-Attention. This post-training strategy modifies the underlying architecture by an attention mechanism that is selectively switched on when the model predicted the disease. They compare their approach to simple self-attention-based architectures on two real-world datasets.

**Strengths:**

-	The idea of incorporating having a label-guided attention seems novel, albeit their use still has to be proven (see Weaknesses)
-	The underlying problem of discovering more interpretable architectures in the biological domain is relevant, but this paper does not seem to make a significant contribution in this direction

**Weaknesses:**

-	There is a general **dissonance between what the paper title suggest as scope** (general interpretable transformers by a new attention mechanism) **versus what it actually shows** (a specific architecture for binary AD classification for single cells)
-	The **method section lacks key details of the architecture and notation**, which makes it not self-contained. The formal introduction is outsourced to Appendix A, yet its notations is relevant to understand the architecture in the main paper (for example, what inputs actually represent), as well as how the evaluation on pathways is carried out. Similarly $C_i$ and $P$ are introduced long after first used, or only in the Appendix.
-	The **resulting model is arguably useful as it reaches a classification performance barely above chance** on ROSMAP data and arguably bad performance on Seattle data given it is only binary classification. This means that any conclusions drawn from this model are arguably relevant for the actual disease, and the guidance signal – which is the *predicted* label – will be close to random noise.
-	The paper **lacks information and analysis of design choices of architecture and training** that could have led to this bad performance (choice of #samples, splits, pathways as input – see Questions)
-	The authors **do not provide a thorough analysis and comparison in the Experiments**: How does the method compare to a model of *equal size* (given that #parameters are almost doubled) and *equal training* (given that there is an additional fine-tuning in terms of number of iterations and samples)? How does the approach compare to SOTA methods such as TOSICA, Geneformer, and scGPT with a classification head for AD classification? Attribution or attention score analysis can be applied to these models in a similar way as presented here. Is the model overfitting to the data including batch effects: How does the model generalize across datasets given the task (AD prediction) stays the same?
-	The study uses the original ROSMAP data which is **known to have severe batch effects biases and artifacts** that influence the results and makes them arguably useful. **A harmonized dataset exists and should be used instead** [1].
-	The **attribution methods used in this paper are outdated** and belong to the first generation of such methods from almost a decade ago. Extension of those methods exist, such as Integrated Gradient 2, as well as approaches that perform much better on benchmarks, such as LRP, making up the second generation. For transformers, within the last few years the third generation has arrived, with CheferLRP [2], LeGrad [3], and T-attn [4], **which instead should be considered here for a fair comparison**.

[1] M Flotho et al. *ROSMAP-Compass: a data-harmonised, AI-ready atlas of 22 million single nuclei from the ROSMAP cohort.* bioRxiv 2025. doi: https://doi.org/10.1101/2025.08.11.668964

[2] H Chefer et al. * Transformer Interpretability Beyond Attention Visualization.* CVPR 2021.

[3] W Bousselham et al. * LeGrad: An Explainability Method for Vision Transformers via Feature Formation Sensitivity.* ICCV 2025 (available on arXiv since 2024).

[4] J Chen et al. *Beyond Intuition: Rethinking Token Attributions inside Transformers.* TMLR 2024.

**Questions:**

Apart from the questions raised above:

-	What is the **justification for modeling by pathways as input features**, would the performance be better using genes instead? What is commonly done is a GSEA or GSOR on the gene sets to identify pathways, here for example based on the attribution scores of each gene, which still allows to get interpretable and measurable effects of pathways in these models
-	Did you **ensure that the train and test split are each balanced** with respect to the label?
-	Why is **only so little data considered** for training (50k) given that millions of cells are available in these datasets?
-	**What is actually interpretable about these transformers**, given that interpretations are given on the input (pathway) level by using attribution, something that could be done for any transformer-based architecture?

---

### Official Review · Reviewer_Liy9 · 2025-10-31

**Soundness:** 1
**Presentation:** 2
**Contribution:** 1
**Rating:** 0
**Confidence:** 4

**Summary:**

This paper proposes a Transformer architecture, called DSCGA for classifying single-cell transcriptomic data. The goal is to improve biological interpretability while maintaining classification accuracy of Transformer based models for single cell data. The method first maps gene expression vectors into tokens representing known biological pathways, uses a single layered transformer block, and then trains directly using supervised downstream objective, based on a readout head. After training (post-hoc), it then introduces a "Disease-Specific Condition Guided Self-Attention" (DSCGA) layer, which is a second attention-like mechanism, that is post-trained on a subset of data to explicitly match a ground-truth list of disease-relevant pathways. The authors claim this method massively improves biological interpretability, reporting an  increase in correctly identified disease-pathways.

**Strengths:**

The paper addresses an important and high-impact problem: improving the domain-specific interpretability (in this case, biological) of Transformer models, rather than just focusing on accuracy.

**Weaknesses:**

The paper's core claims are invalidated by several severe flaws, one of which is fatal to the paper's entire empirical conclusion.

- **Invalid primary claim**   The paper's headline and key result—a ~4.7x improvement in interpretability (e.g., 3.96 $\rightarrow$ 18.98 detected pathways in Figure 3)—is an artifact of a logically broken experimental setup. The proposed, specifically the DSCGA compoent is post-trained using a multi-task objective. As described in Section 3.3.1 and Appendix B.2, this involves adding a "secondary output" ($Output2_i$) trained with a Mean Squared Error (MSE) loss to closely approximate the token relevance according to **ground-truth** biological knowledge encoded into the target vector. Thus the model was explicitly trained to output the correct list of pathways, which means "GWAS-based validation" in Figure 3 is not a validation; it is a report of the model's performance on a regression task it was directly trained for. The paper goes on to compare this result against baselines that were not trained on this task, namey, $Model_{Attentionsteering}$ that is a post-hoc inference-only method. This methodological flaw invalidates the core claim of model's emergent interpretability.


- **Limited Novelty:** The paper's core input representation—mapping a cell's gene vector to a set of pathway-tokens (described in Appendix A.2)—is presented as a novel part of the proposed architecture. This method is the central, defining contribution of the `TOSICA` (Chen et al., 2023) paper. `TOSICA`'s abstract clearly states it "enables interpretable cell type annotation using biologically understandable entities, such as pathways or regulons. The methodology used for token encoding , how to embed a token based on a masked path way representation Hadamard product with gene expression values, is copied **exactly** from TOSICA's methodology without crediting the earlier work properly. While the submissions cites this paper, it fails to mention that much of the methodology for incorporating pathways into the single cell modeling is not novel and only an application/specialization of paper by Chen et al., 2023.   This is a significant overstatement of the submission's novelty and lack of proper discussion of prior related work. This means the only novel part of methodology is the DSCGA component, which carries very little novelty.


-  **Mis-reprsenting literature:** Another one of paper's core motivations is that SOTA models like `scBERT`, `Geneformer`, and `TOSICA` "fall short when it comes to biological interpretability." This is a significant misrepresentation of prior work. More specifically:
    * scBERT's abstract explicitly lists "model interpretability" as a key result, stating its "attention mechanism... naturally provides hints".
    * Geneformer's abstract explicitly highlights that its "attention weights" encode "network hierarchy" in a "completely self-supervised manner."
    * And `TOSICA` *title* is "Transformer for one stop **interpretable** cell type annotation."
Thus, the submission is built on a premise that misrepresents the state of the field. A more fair representation would be to give credit to earlier work's effort to draw interpretable insights from the model, and then clearly compare and contrast their results with existing works.

-  The "dynamic activation" of the DSCGA module is mis-advertised. Equation 4 shows the mechanism is $Att\_{matrix} = Softmax((M\_A + M\_{A2} \cdot C\_i) / \sqrt{d_k})$. The paper defines $C\_i$ not as a learned, dynamic gate, but as the hard-coded 0-or-1 prediction from the base model's prediction.  Therefore, for half of the cells that are predicted to have $C_i  = 0$, eg the control or healthy cells, the new module's output are entirely ignored and no gradients will flow through them. This is a strange, non-differentiable if/then, and overall a very strange design choice that is not motivated.

**Questions:**

Regarding Equation 4, can you justify describing the DSCGA mechanism as "dynamically activated" when the gate $C_i$ is a hard-coded (non-differentiable) value derived from the prediction of a separate, external model ($Model_{SA}$)? I find this design to be very strange and not motivated at all.

---

### Official Review · Reviewer_jPXF · 2025-10-31

**Soundness:** 2
**Presentation:** 3
**Contribution:** 2
**Rating:** 2
**Confidence:** 5

**Summary:**

The paper proposed a biologically-inspired transformer architecture for genomic pathway identification and disease outcome prediction. The basic idea is to encode disease-specific pathway knowledge into the attention matrix of the transformer and employ a conditional code to distinct a set of different diseases.

**Strengths:**

The method provide a more flexible framework for incorporating biological pathway knowledge into the transformer architecture.

**Weaknesses:**

The topic of the paper is highly misleading. One would expect to see a general solution for interpreting transformer models. It turned out to be only a biologically inspired transformer layer for encoding gene profile data. It cannot even be called a "interpretable gene transformer" because only the designed pathway layer is interpretable but the decision pattern of transformers from pathways are still unknown.

The method explicitly require a set of diseases as conditions, which limited the application to closed set problems and severely restricted its usefulness. Firstly, transformer models are expected to used as pretrained foundation models that can fit versatile disease situations after fine-tuning. Secondly, for most disease we have little to no prior pathway knowledge. For those we have prior knowledge, we expect the model to discover new pathways rather than known ones. However, the model did not exhibit ability of discovering new pathways that were not previously encoded in training.

The experiment part had many flaws. The authors claim " none of the existing models, including scGPT, scBERT, and Geneformer, were
designed to directly integrate external biological knowledge". However, they did not used any of them as baseline, but instead used plain transformers that were not pre-trained on biological dataset, which would drop the hidden knowledge obtained from large-scale pre-training dataset and is not the case of current applications. The identical performance of DSGCA (the proposed method) and attention steering on positive condition suggested that the proposed method does not really discover pathway knowledge, but merely switch it on/of with the use of the conditional switch. The high false positive rate on attention steering results suggested that it was over-steered （e.g., no irrelevant pathway encoded), though unknown for DSGCA settings. The proper one should be to encode all pathways (relevant and irrelevant), train with condition/disease labels so that each pathway got different activation weights, and see which ones got activated on the test data. Also, it would be meaningless to compare DSGCA with explainable method when they used plain transformer without pre-training on biological datasets, which would only compare the presence-absence of encoded prior knowledge but not the ability of knowledge discovering.

**Questions:**

N/A

---

### Meta-Review · Area_Chair_Sjiy · 2026-01-07

**Summary:**

Across reviewers, the main concerns are that the paper’s framing and claims overreach relative to what is demonstrated. The method (pathway-tokenization plus a disease-conditioned attention add-on) is only evaluated on binary Alzheimer’s disease classification on two scRNA-seq datasets, which does not support the broad “interpretable transformers” scope. Multiple reviewers also find the architectural choices insufficiently motivated and not clearly more principled than simpler alternatives (eg, adapter-style conditioning over pathway tokens). Most importantly, a reviewer argues the headline interpretability gain is largely an artifact of the evaluation design, calling the core empirical conclusion into question. Finally, the experimental study is viewed as incomplete: missing key ablations/sanity checks showing pathway importance causally influences predictions, limited baseline coverage, and limited evidence of generalization beyond this specific setting.

**Reviewer Concerns:**

### Addressed by rebuttal:
None


### Still outstanding:
- Clarifications on DSCGA formulation / training stages and minor presentation issues (figures, typos) can help, but do not change the core evaluation and scope concerns.
- Validity of interpretability claim: Concerns that the main interpretability metric/result is confounded by the evaluation setup; needs stronger, falsifiable evidence (eg, pathway ablation/shuffling controls, sanity checks against random or mismatched pathways, and demonstrating that highlighted pathways materially change the decision).

- Motivation and novelty: Unclear why the specific extra attention term and disease-condition mechanism is necessary versus simpler conditioning (eg, adapter/prompt-like mechanisms over pathway tokens); “dynamically activated” gating is questioned since it is driven by an external non-differentiable signal.

- Experimental completeness: Limited datasets/tasks and limited baselines for both prediction and interpretability; no convincing demonstration that the approach generalizes to other diseases/modalities or that it improves interpretability without unacceptable accuracy trade-offs.

- Scope mismatch: Title/positioning suggests general interpretable transformers, but the evidence is narrowly specific to binary AD classification with curated pathway catalogs.

**Reviewer Scores:**

likely all remained unchanged

---

### Decision · Program_Chairs · 2026-01-26

Reject